# The structure of a dimeric form of SARS-CoV-2 polymerase

Florian A. Jochheim [1], Dimitry Tegunov[1], Hauke S. Hillen [2,3], Jana Schmitzová [1], Goran Kokic [1], Christian Dienemann[1] & Patrick Cramer [1✉]

The coronavirus SARS-CoV-2 uses an RNA-dependent RNA polymerase (RdRp) to replicate and transcribe its genome. Previous structures of the RdRp revealed a monomeric enzyme composed of the catalytic subunit nsp12, two copies of subunit nsp8, and one copy of subunit nsp7. Here we report an alternative, dimeric form of the enzyme and resolve its structure at 5.5 Å resolution. In this structure, the two RdRps contain only one copy of nsp8 each and dimerize via their nsp7 subunits to adopt an antiparallel arrangement. We speculate that the RdRp dimer facilitates template switching during production of sub-genomic RNAs.

[1] Max Planck Institute for Biophysical Chemistry, Department of Molecular Biology, Göttingen, Germany. [2] Max Planck Institute for Biophysical Chemistry, Research Group Structure and Function of Molecular Machines, Göttingen, Germany. [3] University Medical Center Göttingen, Department of Cellular Biochemistry, Göttingen, Germany. ✉email: pcramer@mpibpc.mpg.de

Replication and transcription of the RNA genome of the coronavirus SARS-CoV-2 rely on the viral RNA-dependent RNA polymerase (RdRp)[1–5]. Following the structure of the RdRp of SARS-CoV[6], structures of the RdRp of SARS-CoV-2 were obtained in free form[7] and as a complex with bound RNA template–product duplex[8–11]. These structures revealed a monomeric RdRp with a subunit stoichiometry of one copy of the catalytic subunit nsp12[12], two copies of the accessory subunit nsp8[13], and one copy of the accessory subunit nsp7[3,14]. Two studies additionally observed monomeric RdRp lacking one of the two nsp8 subunits and nsp7[6,9]. Here we show that the RdRp of SARS-CoV-2 can also adopt a dimeric form with two RdRps arranged in an antiparallel fashion.

## Results and discussion

To detect possible higher-order RdRp assemblies in our cryo-EM data for RdRp–RNA complexes, we wrote a script to systematically search for dimeric particles (Methods). We calculated nearest-neighbor (NN) distances and relative orientations of neighboring RdRp enzymes. In one of our published data sets (structure 3[11]), we detected many particles that showed RdRps with a preferred NN distance of 80 Å and a relative orientation of 180° (Supplementary Fig. 1a, b), indicating the existence of a structurally defined RdRp dimer.

From a total of 78,787 dimeric particles, we selected 27,473 particles that showed a strong RNA signal during 2D classification. The selected particles led to a 3D reconstruction at an overall resolution of 6.1 Å (Supplementary Fig. 2b). We then fitted the obtained density with two RdRp–RNA complexes[10], which revealed an antiparallel arrangement and RNA exiting to opposite sides of the dimeric particle. When we applied the two-fold symmetry during 3D reconstruction, the resolution increased to 5.5 Å (Supplementary Fig. 2c).

The reconstruction unambiguously showed that both RdRps lacked one copy of nsp8, and thus each enzyme was comprised of only one copy of each of the three subunits (Supplementary Fig. 3a). The lacking nsp8 is the one that interacts with nsp7 in monomeric RdRp and had been called nsp8b[10]. The reconstruction showed poor density for the C-terminal helix of nsp7 (residues 63–73) and the sliding pole of the remaining nsp8 subunit (nsp8a, residues 6–110) (Supplementary Fig. 2b). These regions were mobile in both RdRp complexes and were removed from the model. Rigid body fitting of the known RdRp domains led to the final structure.

The structure of the antiparallel RdRp dimer showed that the two polymerases interact via their nsp7 subunits, with the nsp7 helices α1 and α3 (residues 2–20 and 44–62, respectively) contacting each other (Fig. 1). Formation of the nsp7–nsp7 dimer interface is only possible upon dissociation of nsp8b, which liberates the dimerization region of nsp7 (Supplementary Fig. 3c). To our knowledge, no similar nsp7–nsp7 interaction has been observed so far, as it differs from a previously described interaction in a nsp7–nsp8 hexadecamer structure[15] and the nsp7–nsp7 interaction observed in a (nsp7–nsp8)₂ heterotetramer[16,17].

Frequently occurring mutations in the SARS-CoV-2 genome are predicted to influence formation of the RdRp dimer. The nsp12 mutation P323L[18], coevolved with the globally dominating spike protein mutation D614G[19], is often found in severely affected patients[20] and is predicted based on the structure to stabilize nsp12 association with nsp8a[19,21]. In contrast, the nsp7 mutation S25L is predicted to destabilize nsp7 binding to nsp8a[21] and the nsp7 mutation L71F, which is associated with severe COVID-19[22], may destabilize binding of the nsp7 C-terminal region to nsp8b. We speculate that mutations in RdRp subunits can influence the relative stabilities of the RdRp monomer and dimer.

Neither the catalytic sites nor the RNA duplexes are involved in RdRp dimer formation. It is therefore likely that the two RdRp enzymes remain functional within the dimer structure. The two RdRp enzymes in the dimer may thus be simultaneously involved in RNA-dependent processes. Unfortunately, we could not test whether the RdRp dimer is functional because we were unable to purify it. In particular, we attempted to reconstitute the RdRp with a nsp12:nsp8:nsp7 stoichiometry of 1:1:1, but obtained preparations showed again an apparent stoichiometry of 1:2:1 that was observed in previous RdRp structures. Therefore, the functional relevance of the RdRp dimer reported here needs to be established.

We hypothesize that the RdRp dimer is involved in the production of sub-genomic RNA (sgRNA)[23–25]. In this intricate process, positive-strand genomic RNA (gRNA) is used as a template to synthesize a set of nested, negative-strand sgRNAs that are 5′ and 3′ coterminal with gRNA. The obtained sgRNAs are later used as templates to synthesize viral mRNAs. Production of sgRNAs involves a discontinuous step, a switch of the RdRp from an upstream to a downstream position on the gRNA template[26]. These positions contain transcription regulatory site (TRS) sequences[27–29], but it is enigmatic how a single RdRp enzyme could 'jump' between these.

Our dimer structure suggests a model for sgRNA synthesis that extends a recent proposal[30,31] (Fig. 2). In the model, one RdRp of the dimer (RdRp 1) synthesizes sgRNA from the 3′ end of the gRNA template until it reaches a TRS in the template body (TRS-B). Due to the lack of one nsp8 subunit, the dimeric RdRp is predicted to have lower processivity than monomeric RdRp[10,14] and this may facilitate TRS recognition. The viral helicase nsp13 could then cause backtracking of the RdRp[30,31]. Backtracking exposes the 3′-end of the nascent sgRNA, which is complementary to the TRS and may hybridize with another TRS located in the leader (TRS-L) at the 5′-end of the template. The resulting RNA duplex could then bind to the active center of the second RdRp (RdRp 2) to continue sgRNA synthesis.

In our model, it is not the RdRp that switches to a second RNA position, but instead the RNA switches to a second RdRp. After the switch, RdRp 1 may backtrack further, whereas RdRp 2 could move forward until it reaches the 5′-end of the template. These movements occur on the same template but in opposite directions and would be facilitated by the antiparallel arrangement of the polymerases. Superpositions show that only one copy of the template-strand engaged nsp13[32] can be modeled on our dimer structure without clashes (Supplementary Fig. 3d). However, the interaction of this nsp13 copy to the monomeric RdRp is partially mediated by the nsp8 copy that is absent in the dimeric form. Thus, how backtracking in a dimeric complex may be facilitated remains unclear and it is possible that the second nsp13 copy that was previously not observed to be engaged with template RNA is involved in this process (Supplementary Fig. 3e).

Although the functional relevance of the RdRp dimer remains to be established, we note that RdRp dimerization and oligomerization has been reported for many other viruses including Influenza, Polio, Hepatitis C, Norovirus, and others[33]. RdRp oligomerisation can be important for cooperative template binding[34] and can be critical for the viral life cycle[35,36]. Future work should therefore concentrate on the preparation and functional analysis of the coronavirus RdRp dimer and possible additional higher-order structures of the enzyme.

## Methods

**Cryo-EM sample preparation**. We reused the already processed data set 3 from our previous publication[11]. Briefly, RNA sequence for the scaffold used was:
rUrUrU rUrCrA rUrGrC rArCrU rGrCrG rUrArG rGrCrU rCrArU rArCrC
rGrUrA rUrUrG rArGrA rCrCrU rUrUrU rGrGrU rCrUrC rArArU rArCrG

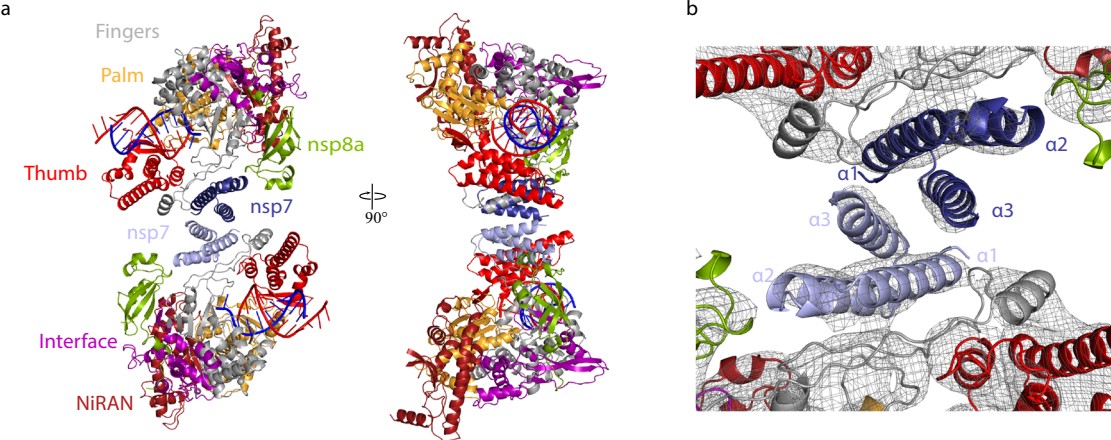

**Fig. 1 Structure of antiparallel RdRp dimer. a** Two views of a ribbon model of the antiparallel RdRp–RNA dimer. Color code for nsp7, nsp8, nsp12 domains (NiRAN, interface, fingers, palm, and thumb), RNA template (blue), and RNA product (red) is used throughout. Nsp7 subunits in the two RdRp monomers are colored slightly differently for the two monomers (dark and light blue, respectively). Views are related by a 90° rotation around the vertical axis. **b** Close-up view of nsp7–nsp7 dimerization interface. View is as in the left structure of panel **a**. The final cryo-EM density is shown as a black mesh.

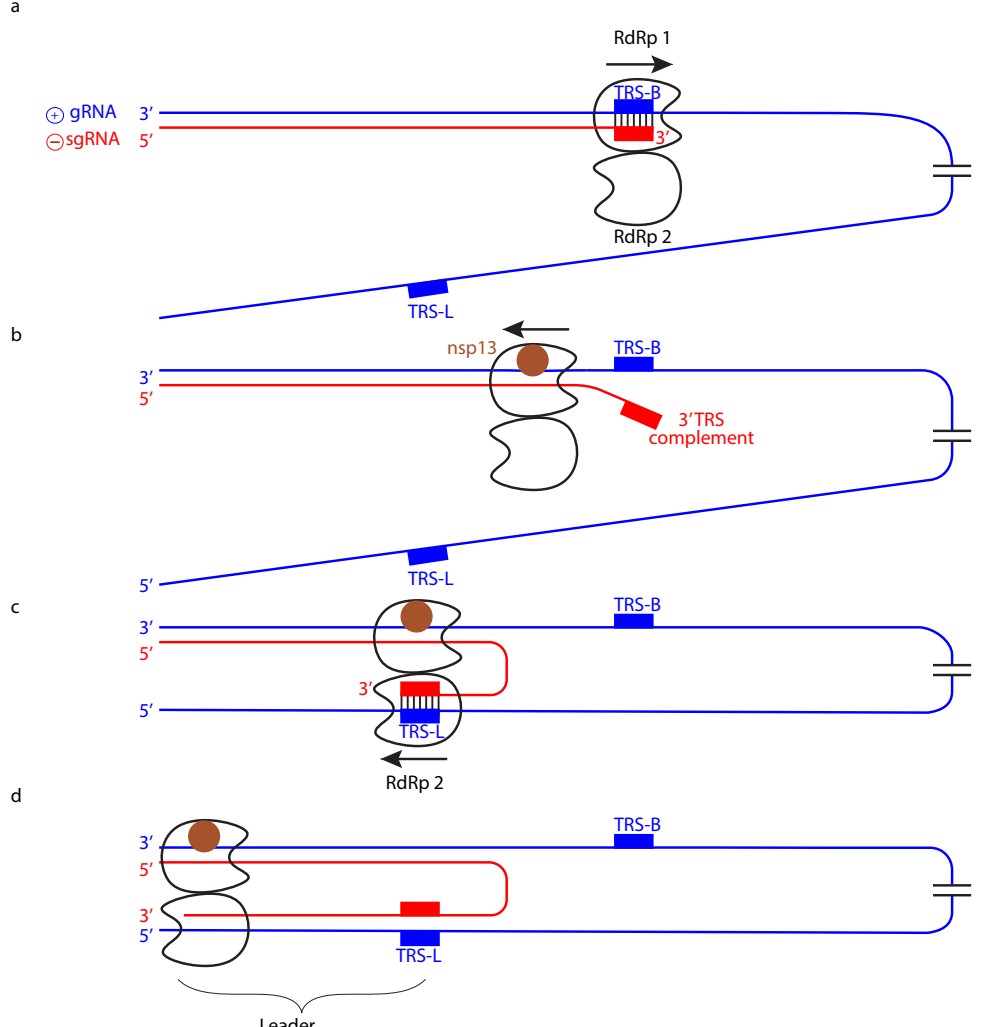

**Fig. 2 Hypothetical model of subgenomic RNA production for viral transcription. a** Genomic positive-strand (⊕) RNA (gRNA) is used as a template to produce the 5′ region of negative-strand (⊖) sgRNA until TRS-B is reached by the RdRp monomer 1. **b** Backtracking is mediated by nsp13 helicase and exposes the newly synthesized, complementary TRS sequence. **c** The complementary sequence in sgRNA can pair with the downstream TRS-L in gRNA and is then loaded into RdRp monomer 2. **d** RdRp 2 then completes ⊖ sgRNA synthesis while RdRp 1 backtracks further.

rGrUrA and rUrGrA rGrCrC rUrArC rGrC- rA/rR -rGrUrG. RdRp–RNA complexes were formed by mixing 1.6 nmol of purified nsp12 with an equimolar amount of RNA scaffold and 4.8 nmol of each nsp8 and nsp7. The mixture was incubated for 10 min and afterwards applied to a Superdex 200 Increase 3.2/ 300 size exclusion chromatography column, which was equilibrated in complex buffer (20 mM Na-HEPES pH 7.4, 100 mM NaCl, 1 mM $MgCl_2$, 1 mM TCEP) at 4 °C. Peak fractions corresponding to the RdRp–RNA complex were pooled and diluted to ~2 mg/ml. Three microliters of the concentrated RdRp–RNA complex were mixed with 0.5 µl of octyl ß-d-glucopyranoside (0.003% final concentration) and applied to freshly glow discharged R 2/1 holey carbon grids (Quantifoil). The grids were blotted for 5 s using a Vitrobot MarkIV (Thermo Fischer Scientific) at 4 °C and 100% humidity and plunge frozen in liquid ethane.

**Preprocessing of cryo-EM data**. Data collection and preprocessing was the same as previously described[11]. Briefly, data was collected using SerialEM[37] on a 300 keV Titan Krios transmission electron microscope (Thermo Fischer Scientific) and a K3 direct electron detector (Gatan). Inelastically scattered electrons were filtered out prior to detection using a GIF quantum energy filter (Gatan) using a slit width of 20 eV. Images were acquired at a nominal magnification of 105,000x and a calibrated pixel size of 0.834 Å/pixel. Due to previously observed preferred orientation when imaging RdRp complexes[10], data was collected using a 30° tilt to obtain more particle orientations. 7043 raw micrographs were acquired in total and preprocessed on-the-fly in Warp for automatic contrast transfer function (CTF) estimation, averaging, motion correction, and particle prediction and extraction. 2.2 million individual RdRp particles were predicted and exported by Warp and imported into cryoSPARC and subjected to a Hetero Refinement job using five 'Junk' classes and one class representing monomeric RdRp as described previously[11]. The resulting particle set was used for a 3D homogeneous refinement to obtain refined poses and positions for each RdRp monomer.

**Initial detection of RdRp dimers in cryo-EM data**. To analyze the statistical distribution of RdRp monomers in our cryo-EM data, we calculated the nearest-neighbor (NN) distances and the relative orientations for all neighboring RdRp complexes using the previously refined monomer poses. To account for the tilted data acquisition, we treated the influence on distances observed on the micrograph as a 30° rotation around the x-axis. We chose to express relative orientation through a single angle by calculating the angle of the rotation around the eigen-vector of the rotation matrix. This showed that certain distances and relative orientations between two monomers were highly prevalent (Supplementary Fig. 1) and indicated the presence of dimeric particles where the two RdRps would adopt a specific distance and relative orientation with respect to each other. We then located such RdRp dimers in micrographs by identifying pairs of RdRps within a narrow range of NN distances and orientations. We observed an overrepresented RdRp distance of 80 Å and relative angle of 180°. Furthermore, we could observe that the overrepresented distance and relative orientation correlated with one another, indicating the occurrence of a defined RdRp dimer (Supplementary Fig. 1a). We initially obtained ~31,000 dimeric particles using a distance <90 Å and relative orientation larger than 166° as a filtering criterion.

**Detection of additional dimeric particles**. Because the yield of dimeric particles depended on both halves of a dimer being first detected as monomer, we aimed to detect more RdRp monomers using two strategies. First, we carried out template-based picking and particle extraction in RELION[38] using the monomeric RdRp–RNA structure[10] filtered to a resolution of 30 Å as a 3D reference to pick further monomers that might have been missed due to their proximity to other particles. Template-based picking of monomeric RdRp however did not introduce any model bias that could influence the dimer structure. Second, we used the previously established NN distance and relative orientation to predict for each RdRp monomer the position where its partner in a dimeric particle should be located on the micrograph. From our NN search we obtained the 3D offset that the second monomer should adopt in a dimeric particle (Supplementary Fig. 1b, c) and used this together with the monomer poses from our first refinement to predict micrograph coordinates for a potential second monomer in a dimeric arrangement. This approach does not bias the analysis towards a fixed relative orientation of NN monomers. Instead, we calculated the orientation of each monomer de novo after combining all picked monomers from Warp, template-based picking, and our prediction approach. With this procedure, patches extracted from predicted micrgraph positions with pure noise will be assigned uncorrelated poses. Only NN monomers that actually form a dimer will have both the correct distance and relative orientation to be predicted as a dimer. After removing duplicates, we used the 3D classification approach described previously[38] and conducted homogenous 3D refinement in cryoSPARC. Using the new monomer populations of 1.5 million particles and their refined poses, we could predict 78,787 dimer particles (i.e., ~10% of the monomers are predicted to be part of a dimer).

**RdRp dimer reconstructions and model building**. Final dimer coordinates were used to extract particles using a box size of 300 Å and a pixel size of 1.201 Å. 2D classification in cryoSPARC[39] (Supplementary Fig. 2a) showed a dimeric structure that contained two RdRps in antiparallel orientation. We selected 2D classes that

showed two RdRp–RNA monomers without a gap between them. After 2D classification, we used 27,473 particles for an ab initio refinement with three classes. For the subsequent 3D homogeneous refinement job in cryoSPARC, we chose a reconstruction from the ab initio refinement that had two clearly resolved RdRp monomers close to one another as a reference. A molecular model was obtained by placing two copies of RdRp–RNA complexes (PDB-7B3D, structure 3 from our previous publication)[11], removal of mobile regions and fitting of domains as rigid bodies was done in UCSF Chimera[40].

**Reporting summary**. Further information on research design is available in the Nature Research Reporting Summary linked to this article.

## Data availability
Coordinates and structure factors for the SARS-CoV-2 RdRp dimer structure have been deposited in the EM data base (EMDB) and Protein Data Bank (PDB) under accession codes PDB-7OYG and EMD-13116, respectively. Raw cryo-EM data were deposited in the EMPIAR database, under accession code EMPIAR-10741. Raw data for nearest neighbor distances and orientations presented in Supplementary Fig. 1 is provided as Supplementary Data 1.

## Code availability
Code for C# programs used to calculate nearest neighbor statistics, predict coordinates of other monomeric RdRp particles, and to create coordinate files for possible dimers is available under https://github.com/cramerlab/RdRp-DimerDetection.

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

## Acknowledgements

P.C. was supported by the Deutsche Forschungsgemeinschaft (EXC 2067/1 39072994, SFB860, SPP2191) and the ERC Advanced Investigator Grant CHROMATRANS (grant agreement No 882357). H.S.H. was supported by the Deutsche Forschungsgemeinschaft (EXC 2067/1 39072994, SFB1190, FOR2848).

## Author contributions

F.A.J. carried out data analysis, assisted by D.T. H.S.H. and G.K. helped with modeling. H.S.H., G.K., J.S. and C.D. helped with structure interpretation. P.C. supervised the project and wrote the manuscript, with input from all authors.

## Funding

## Competing interests

The authors declare no competing interests.
