## [Peer Review File · Communications Biology]

Reviewers' comments:

Reviewer #1 (Remarks to the Author):

The authors report the structure of a SARS-CoV-2 polymerase dimer, and propose a hypothesis for its biological relevance. In addition, the authors provide code which can be repurposed for searching for dimerisation in other cryoEM datasets. The paper is an interesting case study and should be published with the following points addressed.

1. The authors propose an interesting hypothesis for the role of these dimers. However, the authors state that they did not manage to purify the dimer, and they should elaborate on this point. What was attempted and what happened? In light of this, for the benefit of the readers, the authors must discuss at least a few other alternative hypotheses for the emergence of these dimers in the cryo-EM dataset, e.g. as an artefact of cryo-EM specimen preparation conditions. The authors should also clearly state what fraction of all particles in the dataset are found to be dimers. Currently the reader has to infer this from the original publication of the dataset (Kokic et al 2021).

2. For putting their finding in broader biological context, the authors should mention whether RNA polymerase dimers have been observed in other cases. One suggested example is the influenza polymerase (Chang et al 2015, Fan et al 2019).

3. For clarity, the authors should briefly mention the origin of the proteins used for the preparation of the cryo-EM specimen, as described in (Kokic et al 2021), i.e. that the subunits were recombinantly over-expressed in Hi5/Sf9/E.coli/etc cells, assembled on the RNA, and re-purified by gel filtration. Thus, the authors should clearly state in the text that the in vivo relevance of the obtained dimers is not clear at this stage.

4. This study demonstrates the utility of re-processing already acquired cryo-EM datasets. The authors used the same dataset that they previously analysed in (Kokic et al 2021). It would be commendable if the authors made these raw data publicly available on EMPIAR.

5. The authors should provide a section in the Methods detailing their algorithm for finding nearest neighbour distances and orientations, since this may be particularly useful to the cryo-EM community in future. In particular, the authors should make it clear that their data was collected at tilt and explain the approximate fix for the tilt angle in their code. This is important for the reusability of the code.

6. Reconstruction section:

The authors should specify how the data were processed prior to particle extraction (e.g. same as in Kokic et al 2021?), and in which program the particles were extracted.

It is not clear whether ab initio refinement and 3d refinement were also performed in cryoSPARC or not. If so, the type of refinement needs to be specified (Non-uniform, homogeneous, heterogeneous, etc).

The description of the dimer would benefit from showcasing its flexibility or rigidity, for example by cryoSPARC variability analysis, or focused refinement or multi body refinement in RELION, or any other method.

7. Supplementary figure 1:

Add scale bar to panel c

Panels d,e: must also show FSCs for unmasked and for phase randomised maps in each case.

Related to directional FSCs and comment about anisotropy, the authors need to show orientation distributions with/without C2 symmetry, and a quantitative assessment of the severity of the preferential orientation (e.g. efficiency as in Naydenova & Russo 2017 or other metric).

The map provided for review looks substandard, but in this case the distortions due to preferred orientation do not really make a difference to the interpretation. However, is it possible that any views that are not on the C2 axis of the dimer were missed out by the analysis?

The authors claim a relationship between NN distance and relative orientation in the dimers. This should be better illustrated, e.g. with a scatter plot of NN distance vs relative orientation. This may

help explain the initial distance and angle thresholds choice.

In panel b: x and y are in the reference coordinate system of one of the monomer. It would be helpful to sketch the orientation of the axes of this system relative to the particle. Otherwise these directions are meaningless. Also the meaning of the colour bars (number of particles?) needs to be clarified.

8. Supplementary figure 2:

Rather than explaining in which regions the density is weak, the authors should provide a local resolution map, or some other graphical representation of this.

Reviewer #2 (Remarks to the Author):

The manuscript by Jocheim et al. describes a novel oligomeric assembly of the SARS-CoV-2 nsp7-nsp8-nsp12 complex where one of the previously identified nsp8 subunits is missing and its interaction interface with nsp7 has been replaced with the corresponding surface on an identical nsp7, dimerizing the complex. The analysis of the EM data appears to be thorough and the conclusions of the paper are appropriate for a structure of this resolution. My chief concern with the manuscript is whether this protein complex is biologically relevant and not just a spurious protein oligomerization on an exposed hydrophobic surface. Because the dimers represent such a small proportion of the protein population and no biochemical activity can be demonstrated for the complex, it is impossible to conclude a major role for this dimeric complex based on the data presented. The nsp7-nsp7 interface observed has not been observed in any other structural studies of nsp7 that I am aware of and the presented model for nsp13 engagement is inconsistent with this oligomeric assembly (see below). While the structural determination appears to be strong, there is inadequate complementary evidence to justify the paper's conclusions.

On page 2, the authors state that the nsp7-nsp7 interaction is consistent with a (nsp7-nsp8)₂ heterotetramer whose structure was previously determined (reference 16). Superimposition of 7DCD.pdb onto the presented anti-parallel dimer structure indicates that these oligomers are distinct and mutually exclusive using non-overlapping surfaces to make nsp7-nsp7 contacts. Reference 17, only describes heterotrimers and heterotetramers and for SARS-CoV-2 does not suggest a nsp7-nsp7 interface, rather an organization of the oligomers around nsp8.

The discussion of SARS-CoV-2 variant mutations on page 3 is highly speculative. While the effects of several mutations in CoV nsp are predicted none are confirmed to actually influence the interactions of viral nsps experimentally and should not be used to draw further conclusions.

The inability to purify the enzyme to demonstrate that this oligomeric form has activity diminishes the enthusiasm for this work and leaves the question of the biological relevance of this structure unresolved.

In the model discussed on page 4, the authors state that TRS recognition may be facilitated by the dimeric RdRp that has lower processivity due to a missing nsp8 subunit and that this could subsequently allow nsp13 to backtrack the RNA. However, the template-binding nsp13 primarily makes use of the missing nsp8 subunit for its protein interactions and the absence of this nsp8 will likely preclude known interactions with the template-binding nsp13.

In the methods described on page 5, it is not clear how using NN distances and relative orientations of monomeric particles to identify dimeric particles does not bias the analysis to a relative orientation of monomers.

Reviewer #3 (Remarks to the Author):

In this short manuscript, Jocheim et al. report the finding of a dimeric form of the SARS-CoV-2 polymerase complex, arranged in an antiparallel fashion, and discuss its potential involvement in

the conserved discontinuous transcription process of subgenomic RNAs.

The presented results are very interesting, but their interpretation remains speculative, and the proposed model will certainly need to be validated experimentally in the future.

Specific comments

What is the proportion of monomeric and dimeric forms of the RdRp complex?

Have RdRp dimers been observed in other datasets or in RdRp structures from other Coronaviruses?

The paragraph starting with "Frequently occurring mutations in the SARS-CoV-2 genome are predicted" and ending with "can influence the relative stabilities of the RdRp monomer and dimer" contain references about the severity of the clinical outcome of SARS-CoV-2 infections that only vaguely relate to the authors arguments. Do the highlighted mutations impact the formation of RdRp dimers?

Other comments:

It would be very informative to shortly summarize and discuss the experimental conditions employed in reference 11 to obtain structure 3, on which this manuscript is based.

Reviewer #1

1. The authors propose an interesting hypothesis for the role of these dimers. However, the authors state that they did not manage to purify the dimer, and they should elaborate on this point. What was attempted and what happened? In light of this, for the benefit of the readers, the authors must discuss at least a few other alternative hypotheses for the emergence of these dimers in the cryo-EM dataset, e.g. as an artefact of cryo-EM specimen preparation conditions.

We agree and added two sentences to the manuscript that makes this point clear, as follows (lines 72-75):

“In particular, we attempted to reconstitute the RdRp with a nsp12:nsp8:nsp7 stoichiometry of 1:1:1, but obtained preparations that again showed an apparent stoichiometry of 1:2:1 that was observed in previous RdRp structures. Therefore, the functional relevance of the RdRp dimer reported here needs to be established.”

In more detail, we attempted to reconstitute the structurally observed RdRp-RNA complex dimer using different nsp12:nsp8:nsp7 subunit ratios and a RNA template-product duplex. We separated the resulting species by size exclusion chromatography and analysed the obtained peaks by SDS-PAGE. We had previously shown (Hillen et al., Kokic et al.) that when we use a nsp12:nsp8:nsp7 ratio of 1:3:3 for the reconstitution, a ratio of 1:2:1 is observed structurally. When we used instead a subunit ratio of 1:1:1, we obtained a Coomassie-stained SDS-PAGE showing super-stoichiometric amounts of nsp8, suggesting the subunit ratio of the reconstituted RdRp-RNA complex was again 1:2:1.

With some protein-RNA preparations, we could see a peak on the size exclusion column that would correspond to a dimer, but again the apparent stoichiometry of the subunits according to Coomassie staining was 1:2:1. Cryo-EM imaging of the dimeric fraction did not reveal RdRp dimers, suggesting these potentially alternative dimers would not be stable under cryo-EM conditions (Florian Kabinger, unpublished results). In summary, we were not able biochemically to prepare a sample that resemble the RdRp dimer with a 1:1:1 subunit stoichiometry that we structurally defined here.

We also now say that we ‘hypothesize’ that the dimer is involved in function. (line 77)

2. The authors should also clearly state what fraction of all particles in the dataset are found to be dimers. Currently the reader has to infer this from the original publication of the dataset (Kokic et al 2021).

We have added the fraction value to the main text. (line 182)

3. For putting their finding in broader biological context, the authors should mention whether RNA polymerase dimers have been observed in other cases. One suggested example is the influenza polymerase (Chang et al 2015, Fan et al 2019).

We have added a new paragraph at the end of the manuscript with a few examples of RNA polymerase dimers from other viruses and provided the four references below, including the two citations mentioned by the reviewer. (line 106-111)

te Velthuis, A.J.W. Common and unique features of viral RNA-dependent polymerases. *Cell. Mol. Life Sci.* **71**, 4403–4420 (2014). <https://doi.org/10.1007/s00018-014-1695-z>

Högbom M, Jäger K, Robel I, Unge T, Rohayem J. The active form of the norovirus RNA-dependent RNA polymerase is a homodimer with cooperative activity. *J Gen Virol.* 2009 Feb;90(Pt 2):281-291. doi: 10.1099/vir.0.005629-0. PMID: 19141436.

Chang S, Sun D, Liang H, Wang J, Li J, Guo L, Wang X, Guan C, Boruah BM, Yuan L, Feng F, Yang M, Wang L, Wang Y, Wojdyla J, Li L, Wang J, Wang M, Cheng G, Wang HW, Liu Y. Cryo-EM structure of influenza virus RNA polymerase complex at 4.3 Å resolution. *Mol Cell.* 2015 Mar 5;57(5):925-935. doi: 10.1016/j.molcel.2014.12.031. Epub 2015 Jan 22. PMID: 25620561.

Fan H, Walker AP, Carrique L, et al. Structures of influenza A virus RNA polymerase offer insight into viral genome replication. *Nature.* 2019;573(7773):287-290. doi:10.1038/s41586-019-1530-7

4. For clarity, the authors should briefly mention the origin of the proteins used for the preparation of the cryo-EM specimen, as described in (Kokic et al 2021), i.e. that the subunits were recombinantly over-expressed in Hi5/Sf9/E.coli/etc cells, assembled on the RNA, and re-purified by gel filtration.

We have added this description to the methods section. (lines 116-129)

5. Thus, the authors should clearly state in the text that the in vivo relevance of the obtained dimers is not clear at this stage.

We have further clarified this in the main text. (lines 74-75)

6. This study demonstrates the utility of re-processing already acquired cryo-EM datasets. The authors used the same dataset that they previously analysed in (Kokic et al 2021). It would be commendable if the authors made these raw data publicly available on EMPIAR.

We have uploaded the raw data to EMPIAR and included the reference. (line 203)

7. The authors should provide a section in the Methods detailing their algorithm for finding nearest neighbor distances and orientations, since this may be particularly useful to the cryo-EM community in future. In particular, the authors should make it clear that their data was collected at tilt and explain the approximate fix for the tilt angle in their code. This is important for the reusability of the code.

We have added further comments within the code itself and the readme file, and have updated the GitHub repository accordingly. We have added a discussion of the tilt angle to the manuscript. (lines 149-150)

8. Reconstruction section: The authors should specify how the data were processed prior to particle extraction (e.g. same as in Kokic et al 2021?), and in which program the particles were extracted.

It is not clear whether ab initio refinement and 3d refinement were also performed in cryoSPARC or not. If so, the type of refinement needs to be specified (Non-uniform, homogeneous, heterogeneous, etc).

We have added a more detailed description of the process to the reconstruction section. (lines 138-144)

9. The description of the dimer would benefit from showcasing its flexibility or rigidity, for example by cryoSPARC variability analysis, or focused refinement or multi body refinement in RELION, or any other method.

Because the strong preferential orientation mostly eliminates overlapping between parts of a dimer, our independent 3D refinement of monomers is equivalent to multi-body refinement with two bodies. We have added further statistical quantification of the relative movement to the manuscript. CryoSPARC's voxel co-variance analysis is ill-suited for motion analysis. (Supplementary Figure 1)

10. Supplementary figure 1:

Add scale bar to panel c

Panels d,e: must also show FSCs for unmasked and for phase randomized maps in each case.

A scale bar and additional FSC curves have been added. (Now Supplementary Figure 2)

11. Related to directional FSCs and comment about anisotropy, the authors need to show orientation distributions with/without C2 symmetry, and a quantitative assessment of the severity of the preferential orientation (e.g. efficiency as in Naydenova & Russo 2017 or other metric).

We have added an orientation distribution with C2 symmetry, as well as the sphericity score from the 3D-FSC server to quantify the anisotropy. (Now Supplementary Figure 2)

12. The map provided for review looks substandard, but in this case the distortions due to preferred orientation do not really make a difference to the interpretation. However, is it possible that any views that are not on the C2 axis of the dimer were missed out by the analysis?

We had previously tested this by template-matching with a dimer map at all possible angles, but did not find dimers outside of the very limited angular distribution.

13. The authors claim a relationship between NN distance and relative orientation in the dimers. This should be better illustrated, e.g. with a scatter plot of NN distance vs relative orientation. This may help explain the initial distance and angle thresholds choice.

We have plotted NN distance against relative orientation to visualize this and included this information in the revised manuscript. (Supplementary Figure 1, lines 157-159)

14. In panel b: x and y are in the reference coordinate system of one of the monomer. It would be helpful to sketch the orientation of the axes of this system relative to the particle. Otherwise these directions are meaningless. Also the meaning of the colour bars (number of particles?) needs to be clarified.

We have visualized the axes with respect to the dimer and added the missing labels. (Supplementary Figure 1)

15. Supplementary figure 2:

Rather than explaining in which regions the density is weak, the authors should provide a local resolution map, or some other graphical representation of this.

We have added an iso-surface rendering colored by local resolution. (Now Supplementary Figure 3)

Reviewer #2:

16. My chief concern with the manuscript is whether this protein complex is biologically relevant and not just a spurious protein oligomerization on an exposed hydrophobic surface. Because the dimers represent such small proportion of the protein population and no biochemical activity can be demonstrated for the complex, it is impossible to conclude a major role for this dimeric complex based on the data presented.

The key argument why this dimeric complex is not a spurious oligomerization product is the defined nature of the dimer interface, its conformationally defined state, and its stability under cryo-EM conditions. As pointed out in the manuscript, however, we were unable to provide evidence for the biological relevance and have made sure this is clearly understood. Nevertheless, we believe this observation is important for the rapidly growing field of SARS-CoV-2 polymerase structure and function, and will contribute to developing testable hypothesis on the mechanism of coronavirus replication. Therefore, we feel it should be reported in the literature in a timely manner.

17. The nsp7-nsp7 interface observed has not been observed in any other structural studies of nsp7 that I am aware of and the presented model for nsp13 engagement is inconsistent with this oligomeric assembly (see below). While the structural determination appears to be strong, there is inadequate complementary evidence to justify the paper's conclusions.

It is correct that the observed nsp7-nsp7 interaction has not been observed before and this makes our observation novel and worth reporting. With respect to the second point, this may be a misunderstanding. Modeling shows that one copy of nsp13 can bind the dimer, but not the other, which is consistent with our speculative functional model. We made sure this is correctly understood from the revised text. (lines 99-104)

18. On page 2, the authors state that the nsp7-nsp7 interaction is consistent with a (nsp7-nsp8)₂ heterotetramer whose structure was previously determined (reference 16). Superimposition of 7DCD.pdb onto the presented anti-parallel dimer structure indicates that these oligomers are distinct and mutually exclusive using non-overlapping surfaces to make nsp7-nsp7 contacts. Reference 17, only describes heterotrimers and heterotetramers and for SARS-CoV-2 does not suggest a nsp7-nsp7 interface, rather an organization of the oligomers around nsp8.

We went back to the comparison and found the reviewer is correct. We thank the reviewer for noticing our mistake in our submitted manuscript. We rephrased the text accordingly. (lines 55-58)

19. The discussion of SARS-CoV-2 variant mutations on page 3 is highly speculative. While the effects of several mutations in CoV nsp are predicted none are confirmed to actually influence the interactions of viral nsps experimentally and should not be used to draw further conclusions.

It is common practice that one maps known mutations on a structure and offers possible explanations for their effects. We however agree it must be clearly indicated that the interpretation remains speculative and we made sure that this is correctly understood from the revised text. (lines 65-66)

20. The inability to purify the enzyme to demonstrate that this oligomeric form has activity diminishes the enthusiasm for this work and leaves the question of the biological relevance of this structure unresolved.

Please compare our answer to comment 16 from the same reviewer.

21. In the model discussed on page 4, the authors state that TRS recognition may be facilitated by the dimeric RdRp that has lower processivity due to a missing nsp8 subunit and that this could subsequently allow nsp13 to backtrack the RNA. However, the template-binding nsp13 primarily makes use of the missing nsp8 subunit for its protein interactions and the absence of this nsp8 will likely preclude known interactions with the template-binding nsp13.

It is true that the template-binding nsp13.1 copy we modeled not only interacts with the nsp12 thumb, but also with the nsp8T extension and head domains. It is therefore a valid point that the known nsp13 interaction surfaces are absent or altered, which may affect nsp13 binding. We thank the reviewer for this insightful comment and have adjusted our statements in the revised text. (lines 99-104)

22. In the methods described on page 5, it is not clear how using NN distances and relative orientations of monomeric particles to identify dimeric particles does not bias the analysis to a relative orientation of monomers.

Recalculating the poses of the predicted monomers yields an unbiased value for them which yields unbiased relative orientations for NN. We have added further explanation to the text to clarify this and also clarified existing formulations. (lines 174-179)

Reviewer #3:

23. What is the proportion of monomeric and dimeric forms of the RdRp complex?

We have added these values to the text. (line 182)

24. Have RdRp dimers been observed in other datasets or in RdRp structures from other Coronaviruses?

Please compare our response to reviewer 1, comment 3. Briefly, we have added further examples of RNA polymerase dimers from other viruses. (line 106-111)

25. The paragraph starting with “Frequently occurring mutations in the SARS-CoV-2 genome are predicted” and ending with “can influence the relative stabilities of the RdRp monomer and dimer” contain references about the severity of the clinical outcome of SARS-CoV-2 infections that only vaguely relate to the authors arguments. Do the highlighted mutations impact the formation of RdRp dimers?

It is common practice to map known mutations on a structure and offers possible explanations for their effects. We however agree it must be clearly indicated that the interpretation remains speculative and we made sure that this is correctly understood from the revised text. As also pointed out in the text we were unable to make these dimers biochemically and therefore could not test our possible explanations for the effect of the mutations. (lines 74-75 and 77)

26. It would be very informative to shortly summarize and discuss the experimental conditions employed in reference 11 to obtain structure 3, on which this manuscript is based.

We have added a brief description of relevant details from Kokic et al. (reference 3). (lines 116-129)

REVIEWERS' COMMENTS:

Reviewer #2 (Remarks to the Author):

The authors have done an adequate job in addressing my concerns and clarifying their statements.

Reviewer #3 (Remarks to the Author):

The authors addressed the majority of the comments and have made clear that the interpretation of the functional relevance of the dimeric form of the SARS-CoV-2 polymerase remains speculative, mainly because the dimer could not be purified. It will also be highly interesting to investigate the impact of nsp12 or nsp7 mutations and whether they affect the stability or fraction of dimeric RdRp. The discoveries and methods presented in this article can be used in the future to examine the RdRp complexes from other coronaviruses and investigate the function of polymerase dimers during coronavirus infection.

At line 22, please write SARS-CoV instead of SARS-CoV-1.